# Different CprABC amino acid sequences affect nisin A susceptibility in *Clostridioides difficile* isolates

Noriaki Ide[1]ᴼ, Miki Kawada-Matsuo[2,3]ᴼ, Mi Nguyen-Tra Le[2,3], Junzo Hisatsune[3,4], Hiromi Nishi[1], Toshinori Hara[4,5,6], Norikazu Kitamura[4], Seiya Kashiyama[4,5,6], Michiya Yokozaki[4,6], Hiroyuki Kawaguchi[1], Hiroki Ohge[4,7], Motoyuki Sugai[3,4], Hitoshi Komatsuzawa[2,3]*

1 Department of General Dentistry, Hiroshima University Hospital, Hiroshima, Japan, 2 Department of Bacteriology, Hiroshima University Graduate School of Biomedical and Health Sciences, Hiroshima, Japan, 3 Project Research Center for Nosocomial Infectious Diseases, Hiroshima University, Hiroshima, Japan, 4 Antimicrobial Resistance Research Centre, National Institute of Infectious Diseases, Higashi Murayama, Japan, 5 Section of Clinical Laboratory, Division of Clinical Support, Hiroshima University Hospital, Hiroshima, Japan, 6 Division of Laboratory Medicine, Hiroshima University Hospital, Hiroshima, Japan, 7 Department of Infectious Diseases, Hiroshima University Hospital, Hiroshima, Japan

ᴼ These authors contributed equally to this work.

* komatsuz@hiroshima-u.ac.jp

**Data Availability Statement:** All relevant data are within the paper and its Supporting information files.

## Abstract

Clinical isolates of *Clostridioides difficile* sometimes exhibit multidrug resistance and cause diarrhea after antibiotic administration. Metronidazole and vancomycin are often used as therapeutic agents, but resistance to these antibiotics has been found clinically. Therefore, the development of alternative antimicrobial agents is needed. Nisin A, produced by *Lactococcus lactis*, has been demonstrated to be effective against *C. difficile* infection. In this study, we evaluated the susceptibility of 11 *C. difficile* clinical isolates to nisin A and found that they could be divided into 2 groups: high and low susceptibility. Since CprABC and DltDABC, which are responsible for nisin A efflux and cell surface charge, respectively, have been reported to be related to nisin A susceptibility, we investigated the expression of *cprA* and *dltA* among the 11 strains. *cprA* expression in all strains was induced by nisin A, but *dltA* expression was not. The expression levels of both genes did not correlate with nisin A susceptibility in these clinical isolates. To evaluate cell surface charge, we performed a cytochrome C binding assay and found no relationship between charge and nisin A susceptibility. Then, we determined the whole genome sequence of each clinical isolate and carried out phylogenetic analysis. The 11 isolates separated into two major clusters, which were consistent with the differences in nisin A susceptibility. Furthermore, we found common differences in several amino acids in the sequences of CprA, CprB, and CprC between the two clusters. Therefore, we speculated that the different amino acid sequences of CprABC might be related to nisin A susceptibility. In addition, *C. difficile* strains could be divided in the same two groups based on susceptibility to epidermin and mutacin III, which are structurally similar to nisin A. These results suggest that genotypic variations in *C. difficile* strains confer different susceptibilities to bacteriocins.

**Funding:** The author(s) received no specific funding for this work.

**Competing interests:** The authors have declared that no competing interests exist.

## Introduction

*Clostridioides difficile* is an obligatory anaerobic, spore-forming, gram-positive bacterium. This bacterium produces toxins, toxin A (TcdA), toxin B (TcdB), or binary toxin CDT [1, 2]. In particular, toxin A and toxin B have been demonstrated to be significantly involved in the onset of diarrhea; thus, this bacterium causes colitis related to antibiotic administration [3]. Since some clinical *C. difficile* strains exhibit multidrug resistance, and the administration of several antibiotics, such as β-lactams and quinolones, sometimes disrupts the composition of normal gut microbiota is then followed by the proliferation of *C. difficile* [4, 5]. Therefore, this disease is mostly found in health care-associated infections. Metronidazole and vancomycin have generally been used to treat *C. difficile* infection, but recently, resistance to these antibiotics has been found clinically [4]. Similar to *C. difficile*, many drug-resistant gram-positive and gram-negative bacteria have significantly challenged the available therapeutic drugs [6]. Therefore, actions against antimicrobial resistance (AMR) have been performed around the world. In the 2019 report of antibiotic resistance threats in the US by the Centers for Disease Control and Prevention (CDC), *C. difficile* was ranked as an urgent threat, together with carbapenem-resistant Acinetobacter/Enterobacteriaceae [7]. Therefore, the development of new antibacterial agents against *C. difficile* infection is needed.

Bacteriocins are considered to be new candidate antibiotic agents [8, 9]. Bacteriocins are peptides or proteins that are ribosomally synthesized in some bacterial species, including gram-positive and gram-negative bacteria [10, 11]. Bacteriocins are mainly classified into 2 groups, classes I and II [12, 13]. Class I bacteriocins are post-translationally modified peptides that include the unusual amino acids lanthionine and β-methyllanthionine. Class I bacteriocins include lantibiotics, sactibiotics, linaridins, thiopeptides, glycocins, circular peptides, and bottromycins from gram-positive bacteria and nucleotide peptides and siderophore peptides from gram-negative bacteria [13]. Class II bacteriocins consist of unmodified peptides [14]. Among the many bacteriocins, nisin A produced by *Lactococcus lactis* has been well characterized and is widely used as a food additive worldwide [15, 16]. Nisin A binds to lipid II on the cell membrane and causes pore formation, resulting in bacterial death [12, 17]. Nisin A also inhibits cell wall peptidoglycan biosynthesis because lipid II is associated with its biosynthesis. Therefore, many investigations regarding the effects of nisin A on pathogenic bacteria, including *Staphylococcus aureus*, *C. difficile* and *Clostridium perfringens*, have been performed [18–22]. However, gram-positive bacteria, including *S. aureus*, *S. mutans* and *C. difficile*, have a two-component system (TCS)-mediated factor for resistance to nisin A [23–27]. In *Clostridioides difficile*, CprRK is a TCS that regulates the expression of CprABC, an ABC transporter responsible for nisin A resistance [26, 27]. CprK senses nisin A, which is followed by activation of CprR. In addition, it was reported that the Dlt system, which is associated with cell surface charge, was involved in resistance to cationic antimicrobial agents, including nisin A [28].

In this study, we investigated the susceptibility of 11 clinically isolated *C. difficile* strains to nisin A and found that they could be mainly divided into 2 groups: high and low susceptibility. To elucidate their differences, we performed whole-genome analysis and compared the high- and low-susceptibility strains.

## Materials and methods

### Bacterial strains and culture medium

The *C. difficile* strains used in this study are listed in Table 1. These strains except JCM1296$^T$ and JCM5243 were isolated from patients at Hiroshima University Hospital. *C. difficile* JCM1296$^T$ (NCTC 11209) and JCM5243 were obtained from the Japan Collection of

**Table 1. Strains used in this study.**

| Strains | Character | Reference |
|---|---|---|
| *Clostridioides difficile* JCM5243 | Wild type, laboratory strain, ST[1]3 | Riken BRC [2] |
| *Clostridioides difficile* JCM1296[T] | Wild type, laboratory strain, ST3 | Riken BRC |
| *Clostridioides difficile* H39 | Clinical isolate, ST109 | This study |
| *Clostridioides difficile* H139 | Clinical isolate, ST48 | This study |
| *Clostridioides difficile* H153 | Clinical isolate, ST8 | This study |
| *Clostridioides difficile* H156 | Clinical isolate, ST81 | This study |
| *Clostridioides difficile* H176 | Clinical isolate, ST203 | This study |
| *Clostridioides difficile* H197 | Clinical isolate, ST183 | This study |
| *Clostridioides difficile* H299 | Clinical isolate, ST81 | This study |
| *Clostridioides difficile* H329 | Clinical isolate, ST39 | This study |
| *Clostridioides difficile* H333 | Clinical isolate, ST109 | This study |
| *Lactococcus lactis* ATCC11454 | Nisin A-producing strain | 26 |
| *Lactococcus lactis* QU5 | Lacticin Q-producing strain | 30 |
| *Staphylococcus epidermidis* KSE56 | Epidermin-producing strain | 31 |
| *Staphylococcus epidermidis* KSE650 | Nukacin KSE650-producing strain | 31 |
| *Streptococcus mutans* KSM157 | Mutacin I-producing strain | 32 |
| *Streptococcus mutans* KSM2 | Mutacin III-producing strain | 32 |
| *Streptococcus mutans* KSM170 | Mutacin IIIb-producing strain | 32 |
| *Streptococcus mutans* KSM13 | Mutacin IV-producing strain | 32 |
| *Enterococcus mundtii* QU2 | Mundticin KS-producing strain | 33 |

[1]. Sequence type

[2]. Japan Collection of Microorganisms.

Microorganisms (JCM). The *C. difficile* strains were cultured in brain heart infusion (BHI) broth (Becton, Dickinson and Company, New Jersey, USA) at 37˚C under anaerobic conditions using the GasPack system (Mitsubishi Gas Chemical Company Inc., Tokyo, Japan). All fresh media were kept under anaerobic conditions for at least 5 hours. *Lactococcus lactis* (ATCC11454 [25], QU5 [29]), *Staphylococcus epidermidis* (KSE56, KSE650 [30]), *Streptococcus mutans* (KSM2, KSM13, KSM157, KSM170 [31]) and *Enterococcus mundtii* QU2 [32], were grown aerobically in trypticase soy broth (TSB) at 37 ˚C with (*Streptococcus*, *L. lactis* and *Enterococcus*) or without (*S. epidermidis*) 5% $CO_2$.

## Ethics

All the clinical *C. difficile* isolates were anonymized prior to obtaining by us, and there was no information about patients. The Ethical Committee for Epidemiology of Hiroshima University reviewed our application and approved it (E2019-1941).

## Susceptibility test

Two methods were used for the evaluation of antibacterial activity. To measure the MIC value of nisin A (Sigma–Aldrich, St. Louis, USA), the microdilution method was used as described elsewhere [23]. Bacitracin (Fujifilm Wako chemicals, Osaka, Japan), vancomycin (Sigma–Aldrich), Metronidazole (Fujifilm Wako chemicals), ampicillin (Nacalai Tesque, Kyoto, Japan), chloramphenicol (Wako chemicals), gentamicin (Nacalai Tesque), ofloxacin (Sigma–Aldrich) and imipenem (Fujifilm Wako chemicals) were also used for MIC evaluations.

To assess the antibacterial activity of the bacteriocins, a direct assay was performed by a method described elsewhere [23]. An overnight culture (3 μl) of the bacteriocin-producing strain, as indicator bacteria, was spotted on a TSA plate and cultured at 37°C for 24 h. Then, 3.5 ml of prewarmed BHI soft agar (0.75%) containing *C. difficile* cells ($10^8$ cells/ml) was poured over the TSA plate. The plates were incubated anaerobically at 37°C for 24 h. Then, the diameters of the growth inhibitory zones were measured in three directions. Three independent experiments were performed, and the average diameters were calculated.

## Cytochrome c binding assay

A cytochrome c binding assay was performed to evaluate the cell surface charge as described elsewhere [33]. Bacterial cells in the exponential phase were collected and suspended in 10 mM sodium phosphate buffer (pH 6.8) by adjusting to a concentration of $10^9$ cells/ml. Cytochrome c (Sigma Aldrich) was added to the bacterial suspensions at a final concentration of 250 μg/ml. After 10 min of incubation at room temperature, each bacterial suspension was centrifuged at 15 000 rpm for 5 min. The absorbance of the supernatant at 530 nm was measured. By comparing the absorbance values with and without bacterial cells, the absorption ratio was calculated to reflect the bacterial surface charge.

## Gene expression analysis by quantitative PCR

Quantitative PCR (qPCR) was performed to investigate the expression of genes involved in bacteriocin susceptibility. Overnight cultures (500 μl each) of the *C. difficile* strains were inoculated into 5 ml of BHI broth and then grown at 37 °C under anaerobic conditions. After 5 h (0.2 by optical density at 660 nm), nisin A (final concentration: 16 μg/ml) was added to each bacterial culture. After 30 min, the bacterial cells were collected. RNA extraction, cDNA synthesis and qPCR were performed as described previously [23]. The primers used in this study are listed in S1 Table. The expression of each gene was quantified against 16S rRNA expression.

## Effect of nisin A on germination

Prior to the germination assay, spores were purified by a method described elsewhere [34]. To evaluate the effect of nisin A on germination, we selected 2 strains with different DNA type, JCM5243 (Group 1) and H39 (Group 2). *C. difficile* JCM5243 and H39 were grown in BHI broth with 0.5% yeast extract and 0.1% L-cysteine at 37 °C under anaerobic conditions. The overnight culture was then plated on BHI agar with 0.1% L-cysteine and anaerobically incubated at 37 °C. After 7 days of incubation, the plate was placed at 4 °C for 24 h. Bacterial cells were scraped from the agar plate and washed with phosphate-buffered saline (PBS) 2 times. Then, the cells were suspended in PBS containing 125 mM Tris, 200 mM EDTA and 0.3 mg/ml proteinase K and incubated aerobically with gentle agitation at 37 °C for 2 h. Then, the spores were washed with sterilized water 10 times. The spore solution was stored for up to 1 year at 4 °C prior to use. We confirmed the number of spores in each experiment before use.

Before the germination assay, spore stocks were incubated at 65°C for 30 min. Spores ($10^5$ spores) were added to BHI broth with or without 0.1% taurocholate (an inducer for germination) in the presence or absence of nisin A (32 or 512 μg/ml). The spore solution was incubated anaerobically at 37°C for 30 min. Then, appropriate dilutions were plated on BHI agar with 0.1% taurocholate and incubated anaerobically at 37°C for 2 days. Colony-forming units (CFU) were counted. Three independent experiments were performed.

## Genome sequence analysis of the *C. difficile* isolates

To perform whole-genome sequencing of the *C. difficile* isolates, DNA was extracted from each isolate. Genomic DNA was purified from the lysate using AMPure XP beads (Beckman Coulter, USA). DNA libraries were prepared as described previously [35]. Whole-genome sequencing (WGS) of *C. difficile* isolates was performed using the Illumina MiSeq sequencing platform, followed by annotation with Rapid Annotation using Subsystem Technology (RAST) version 2.0 [36]. A phylogenetic tree was constructed using the CSI Phylogeny 1.4 pipeline available from the Center for Genomic Epidemiology (Lungby, Denmark) for SNP calling and then annotated using the iTOL web-based tool [37]. The tree was drawn to scale, with branch lengths in the same units as those of the evolutionary distances used to infer the phylogenetic tree.

We also compared the 11 clinical genomes with 50 *C. difficile* genomes that are available in the NCBI database. Fifty genomes were randomly selected from the database. Then, we performed phylogenetic tree analysis and compared the amino acid sequences of CprA, B and C among the 61 genomes, including 11 genomes sequenced in this study.

## Accession number

The genome data of the *C. difficile* isolates used in this study have been deposited into the DDBJ Sequence Read Archive (DRA) accession number DRA014785 under the BioProject accession no. PRJDB14281.

## Results

### Susceptibility of *C. difficile* to nisin A

The susceptibility to nisin A varied among the 11 clinical *C. difficile* isolates (Table 2). The MIC values ranged from 2 to 64 μg/ml. Since the final nisin A concentration used in this study was 2.5%, the calculated MIC range was 0.05 to 1.6 μg/ml. The MIC value for each isolate correlated with the results of the direct assays, as the isolates with higher MIC values showed lower inhibitory zones (Table 2, S1 Fig).

Based on phylogenetic tree analysis, the 11 isolates were divided into 2 clusters designated Groups 1 and 2 (Fig 1). Group 1 included 6 isolates, showing nisin A MICs of 8 to 64 μg/ml,

**Table 2. Susceptibility to nisin A by MIC and direct assay method.**

|  | DNA Group | MIC (μg/ml) | Halo size[1] (mm) |
|---|---|---|---|
| JCM1296[T] | 1 | 16 | 16.6 ± 0.00 |
| JCM5243 | 1 | 32 | 15.5 ± 1.19 |
| H139 | 1 | 16 | 16.4 ± 0.99 |
| H153 | 1 | 8 | 16.9 ± 0.14 |
| H176 | 1 | 64 | 15.3 ± 0.31 |
| H197 | 1 | 32 | 15.8 ± 0.85 |
| H39 | 2 | 4 | 17.8 ± 0.65 |
| H156 | 2 | 2 | 19.7 ± 2.83 |
| H299 | 2 | 8 | 17.7 ± 0.50 |
| H329 | 2 | 4 | 19.9 ± 1.98 |
| H333 | 2 | 2 | 20.0 ± 1.59 |

[1]. diameter of inhibition zone.

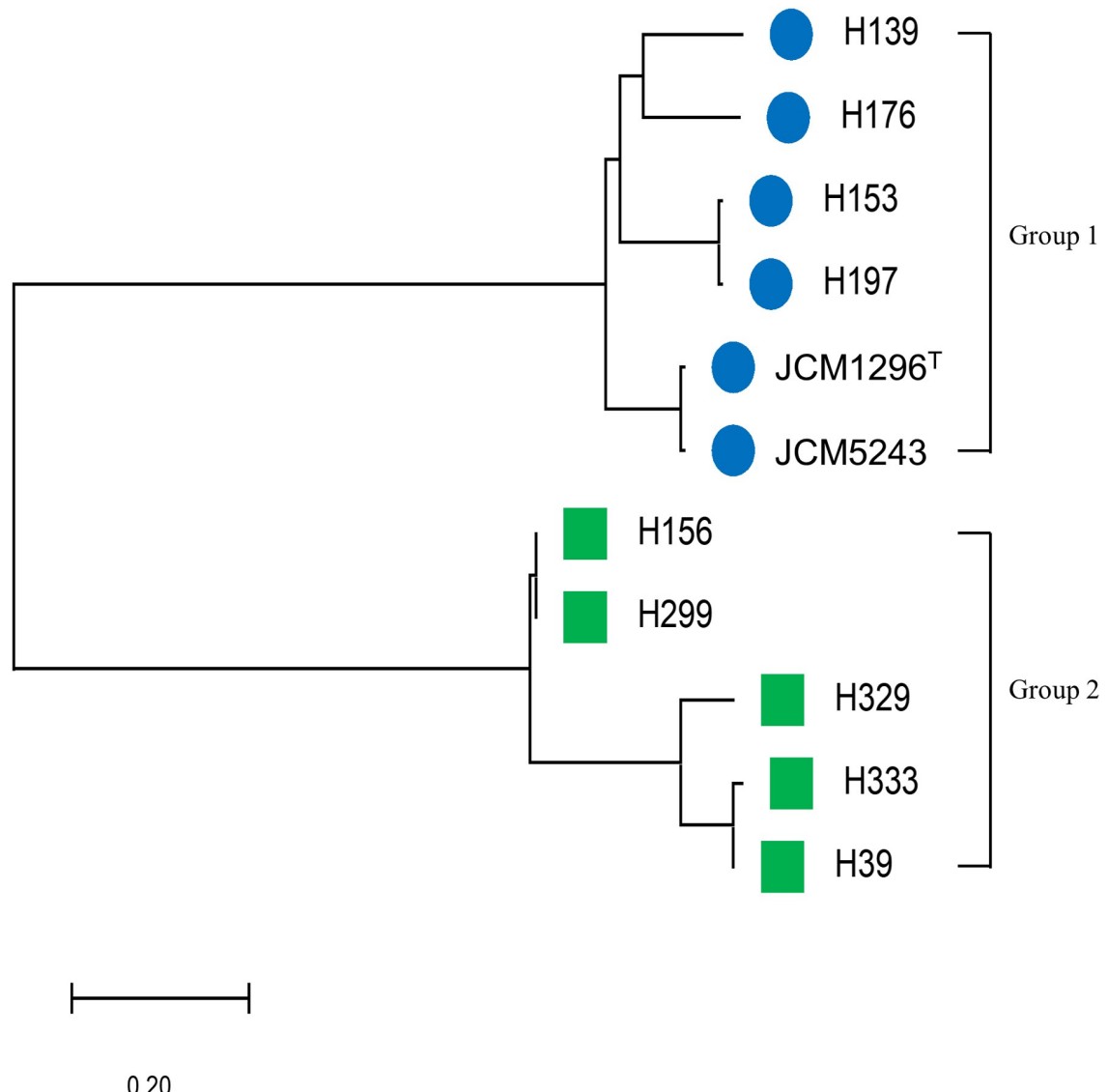

**Fig 1. Phylogenetic analysis of 11 *C. difficile* strains.** A phylogenetic tree was constructed with 11 genome sequences by the method described in Materials and Methods.

while Group 2 included 5 isolates, showing nisin A MICs of 2 to 8 μg/ml (Fig 2A). In the direct assay, the diameter length of Group 1 isolates showed smaller than that of Group 2 isolates (Fig 2B). In both assays, the susceptibility of the strains to nisin A in Group 1 was lower than that of Group 2.

## Susceptibility of *C. difficile* strains to several bacteriocins and antibiotics

Since *C. difficile* displayed susceptibility to nisin A, we evaluated its susceptibility to several lantibiotics and non-lantibiotics (Table 3, Fig 3). The two groups divided by nisin A susceptibility also showed a significant difference in epidermin susceptibility ($p = 0.0081$, Fig 3). Susceptibility against mutacin I ($p = 0.027$), mutacin III ($p = 0.024$) and mutacin IIIb ($p = 0.025$) also tended to be higher in Group 2 than in Group 1. In contrast, no significant difference in

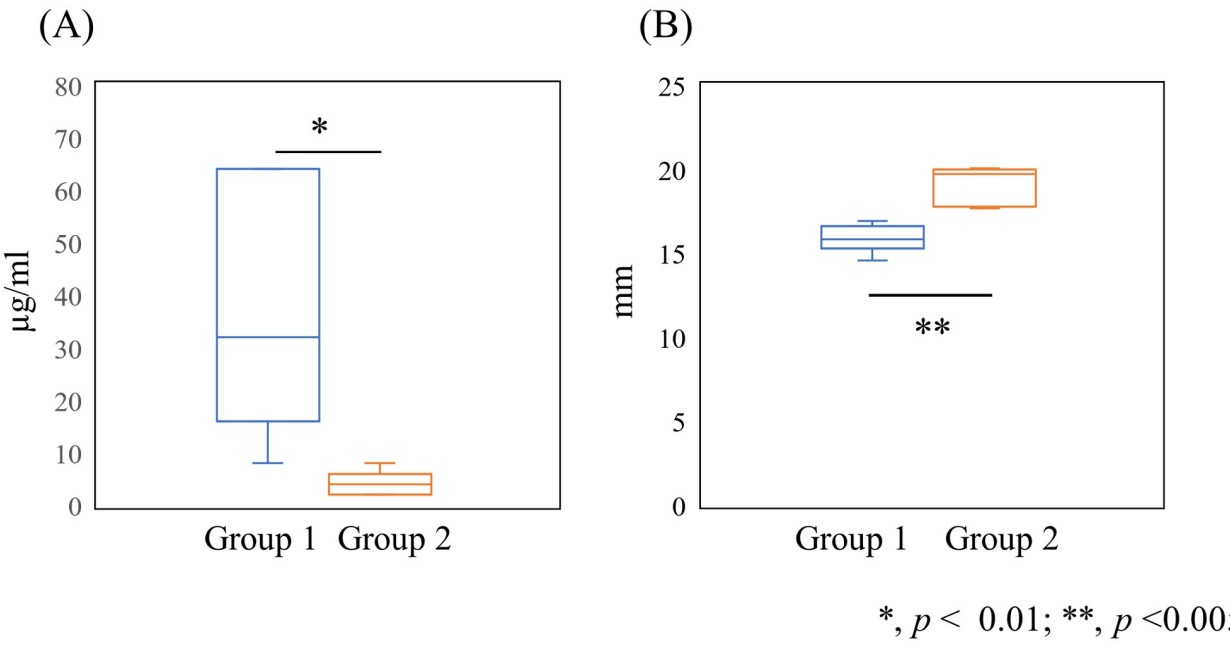

**Fig 2. Comparison of susceptibility to nisin A in Groups 1 and 2.** Susceptibility to nisin A between Group1 and Group 2 was compared by (A) MIC method and (B) the direct assay.

susceptibility to mutacin IV ($p = 0.35$) was found between the two groups. Mundticin KS and lacticin Q showed no antibacterial activity of 9 strains and 7 strains among 12 strains, and we found no significant differences between the two groups (mundticin KS; $p = 0.31$, lacticin Q; $p = 0.21$). In addition, nukacin ISK-1 showed no antibacterial activity against all *C. difficile*

**Table 3. Susceptibility to various bacteriocin-producing strains by direct assay.**

| Strains | DNA Group | Epidermin[1] | Mutacin I[2] | Mutacin III[3] | Mutacin IIIb[4] | Mutacin IV[5] | Mundticin KS[6] | Lacticin Q[7] |
|---|---|---|---|---|---|---|---|---|
| | | | | Halo size (mm) | | | | |
| JCM 1296[T] | 1 | 16.5 ± 0.08 | 9.4 ± 0.04 | 10.7 ± 0.45 | 14.7 ± 0.83 | 6.2 ± 0.05 | 0 | 0 |
| JCM 5243 | 1 | 15.7 ± 0.57 | 9.5 ± 0.08 | 11.5 ± 0.16 | 15.1 ± 0.22 | 6.3 ± 0.37 | 0 | 0 |
| H139 | 1 | 16.7 ± 0.61 | 9.5 ± 0.00 | 13.1 ± 0.22 | 17.4 ± 0.37 | 9.5 ± 0.79 | 0 | 15.6±0.1 |
| H153 | 1 | 18.9 ± 0.37 | 8.8 ± 0.08 | 11.8 ± 0.14 | 15.9 ± 0.28 | 6.4 ± 0.50 | 0 | 13.1±0.3 |
| H176 | 1 | 16.5 ± 0.29 | 10.6 ± 0.16 | 12.4 ± 0.28 | 16.4 ± 0.42 | 6.4 ± 0.08 | 0 | 0 |
| H197 | 1 | 18.2 ± 0.51 | 10.0 ± 0.37 | 11.8 ± 0.54 | 15.8 ± 0.22 | 6.4 ± 0.51 | 10.6±0 | 0 |
| H39 | 2 | 17.6 ± 0.38 | 9.7 ± 0.43 | 11.7 ± 0.57 | 16.3 ± 0.37 | 6.4 ± 0.08 | 0 | 0 |
| H156 | 2 | 19.0 ± 0.00 | 11.1 ± 0.22 | 12.9 ± 0.08 | 16.4 ± 0.59 | 7.4 ± 0.16 | 0 | 7.8±0 |
| H299 | 2 | 24.1 ± 0.43 | 12.4 ± 1.02 | 15.5 ± 0.51 | 19.6 ± 1.77 | 7.1 ± 0.16 | 0 | 19.6±0.06 |
| H329 | 2 | 20.2 ± 0.42 | 10.1 ± 0.22 | 13.0 ± 0.99 | 17.2 ± 0.26 | 7.1 ± 0.08 | 10.5±0 | 0 |
| H333 | 2 | 20.0 ± 0.91 | 10.1 ± 1.14 | 13.2 ± 0.78 | 16.9 ± 0.65 | 7.1 ± 0.16 | 12.7±0 | 13.8±0 |

[1], *Staphylococcus epidermidis* KSE56

[2], *Streptococcus mutans* KSM157

[3], *Streptococcus mutans* KSM2

[4], *Streptococcus mutans* KSM170

[5], *Streptococcus mutans* KSM13

[6], *Enterococcus mundtii* QU2

[7], *Lactococcus lactis* QU5

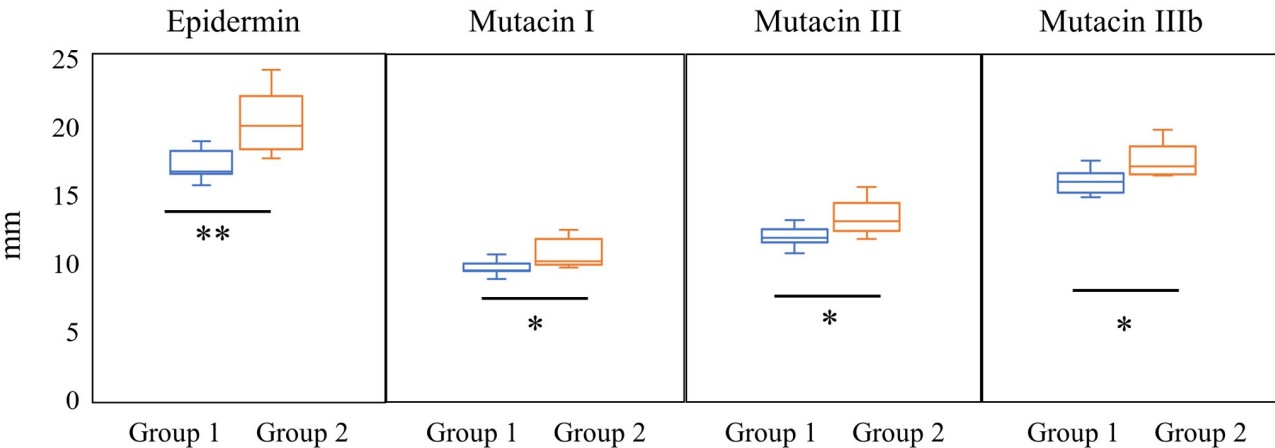

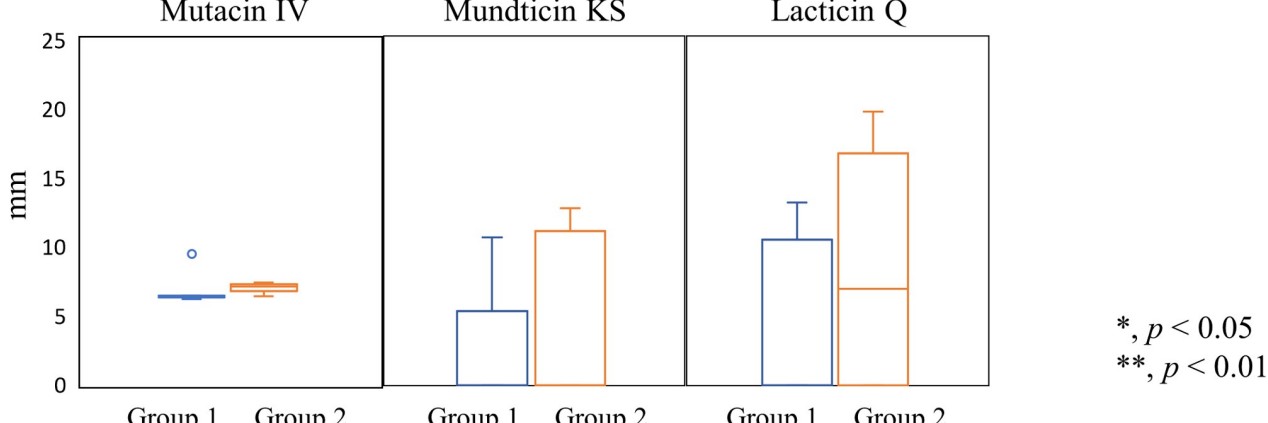

**Fig 3. Comparison of susceptibility to epidermin, and mutacin I/II/III in Groups 1 and 2.** The distribution of the inhibitory zones (diameter: mm) are shown. Student's t test was used to determine significant differences between the Group 1 and Group 2 *C. difficile* strains.

strains. Therefore, lantibiotics with structurally similar to nisin A showed a similar pattern of the susceptibility to nisin A among *C. difficile* isolates.

Then, we further investigated the susceptibility of *C. difficile* strains to several antimicrobial agents (Table 4, Fig 4). However, susceptibility of Groups 1 and 2 to the other antibacterial agents showed no significant differences.

### Expression of *cprA* and *dltD* in *C. difficile* strains

Since *cprA-C* and *dltA-D* were previously reported to be associated with nisin A resistance [26–28], we investigated the expression of *cprA* and *dltD* in the strains with or without nisin A addition (Fig 5). All isolates showed that *cprA* expression was significantly induced by the addition of sub-MICs of nisin A, although the expression levels varied among the strains. The expression level in each strain was not separated by the groups classified defined by nisin A

**Table 4. MIC of various antibiotics.**

| Strains | DNA Group | MIC (µg/ml) | | | | | | | |
|---|---|---|---|---|---|---|---|---|---|
| | | AMP[1] | CP[2] | GM[3] | IP[4] | MNZ[5] | OFLX[6] | BC[7] | VC[8] |
| JCM1296[T] | 1 | 2 | 8 | 256 | 4 | 0.5 | 8 | 512 | 1 |
| JCM5243 | 1 | 1 | 2 | 128 | 4 | 16 | 8 | 512 | 2 |
| H139 | 1 | 0.25 | 1 | 128 | 2 | 0.5 | 8 | 256 | 1 |
| H153 | 1 | 2 | 1 | 128 | 4 | 0.25 | 32 | 256 | 1 |
| H176 | 1 | 0.5 | 1 | 128 | 2 | 0.25 | 8 | 256 | 0.5 |
| H197 | 1 | 0.5 | 1 | 1024 | 4 | 0.125 | 16 | 64 | 2 |
| H39 | 2 | 0.5 | 2 | 512 | 4 | 16 | 8 | 256 | 0.5 |
| H156 | 2 | 1 | 2 | 64 | 4 | 16 | 32 | 512 | 2 |
| H299 | 2 | 4 | 1 | 64 | 2 | 0.25 | 32 | 256 | 0.5 |
| H329 | 2 | 0.5 | 1 | 64 | 1 | 0.125 | 32 | 256 | 0.25 |
| H333 | 2 | 1 | 1 | 64 | 4 | 0.125 | 8 | 32 | 0.5 |

[1], Ampicillin

[2], Chloramphenicol

[3], Gentamicin

[4], Imipenem

[5], Metronidazole

[6], Ofloxacin

[7], Bacitracin

[8], Vancomycin

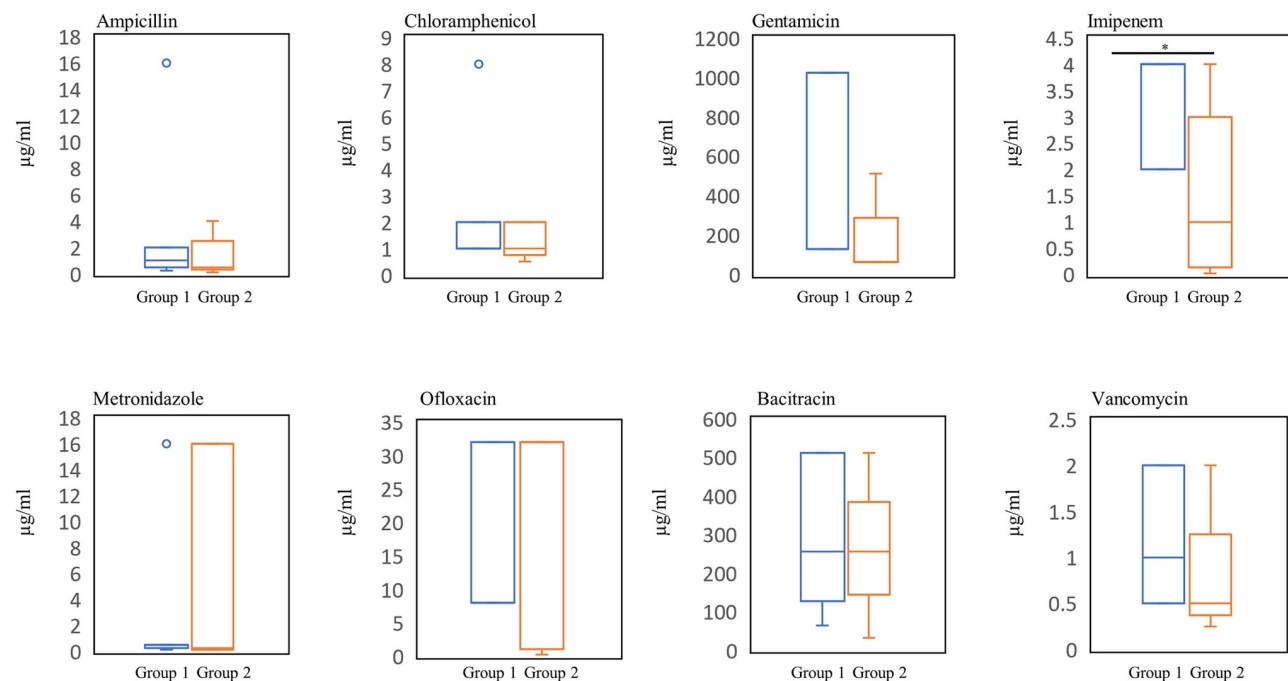

**Fig 4. Comparison of susceptibility of Groups 1 and Group 2 to antibacterial agents.** The distribution of MICs is shown. Student's t test was used to determine significant differences between the Group 1 and Group 2 *C. difficile* strains.

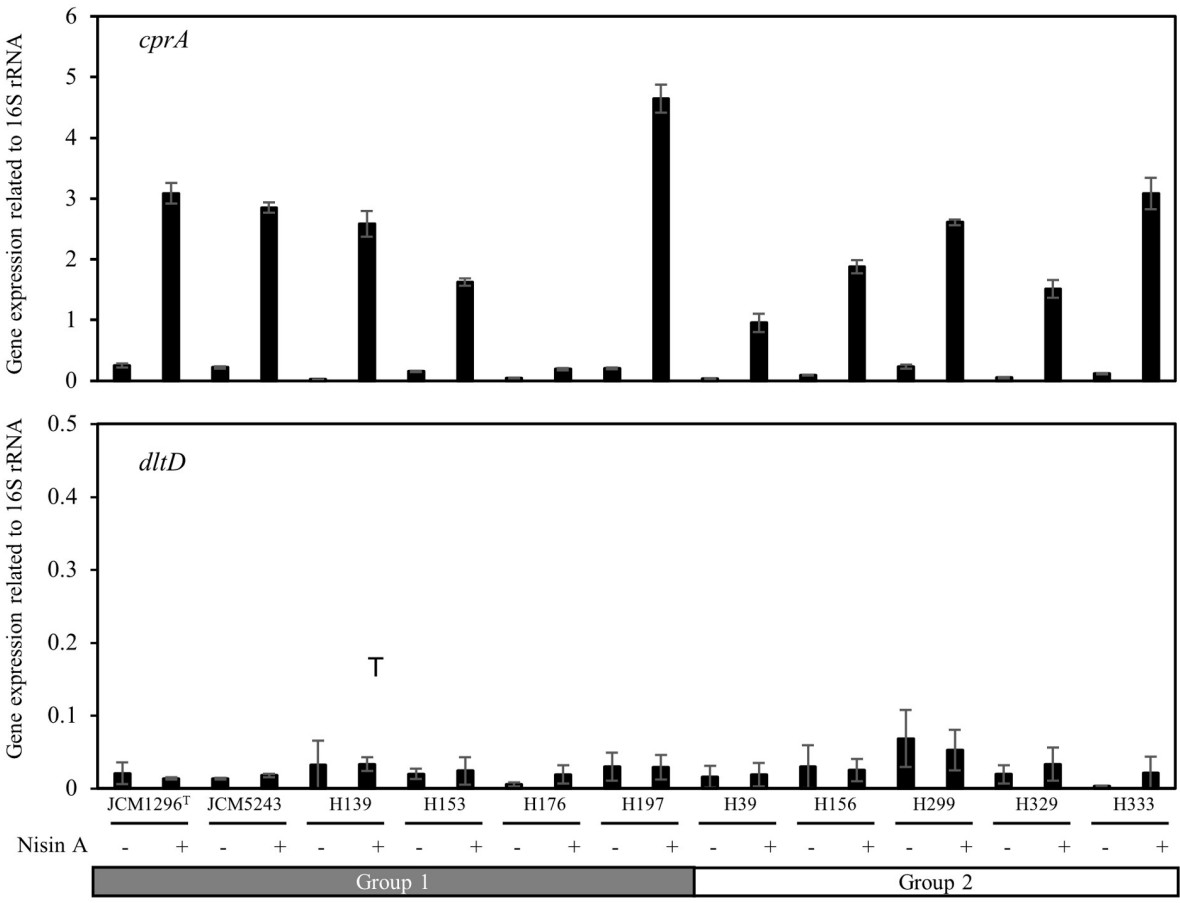

**Fig 5. *cprA* and *dltD* expression among the 11 *C. difficile* strains.** *cprA* and *dltD* expression in 11 *C. difficile* strains with or without nisin A treatment was investigated. Total RNA was extracted from cells in the exponential phase. After cDNA synthesis, quantitative PCR was performed by using specific primers for *cprA* and *dltD*. *16S rRNA* was used as an internal control.

susceptibility. In the absence of nisin A, the *cprA* expression level was quite low in all strains. *dltD* expression was not induced by the addition of nisin A, and was very low in both the presence and absence of nisin A. The expression level of *dltD* among the 11 isolates showed no correlation with grouping.

## Cytochrome c binding assay

We further investigated the cell surface charge because cell surface charge was associated with the susceptibility to cationic peptides including nisin A [28]. To evaluate the cell surface charge, a cytochrome binding assay was performed (Fig 6). Binding affinity was similar among all strains. There was no relationship between the binding affinity and nisin A susceptibility.

## Comparison of CprA-C amino acid sequences

Since we found no correlation between nisin A susceptibility and the expression of *cprA* and *dltD*, we compared the amino acid sequences of CprA-C, which are intrinsic factors for nisin A resistance, among the strains (Fig 7). We found some consensus differences in several amino acids of CprA-C between Groups 1 and 2. The amino acid residues of CprA at positions 30 (serine), 70 (threonine), 116 (threonine) and 193 (glutamic acid) in Group 1 were different

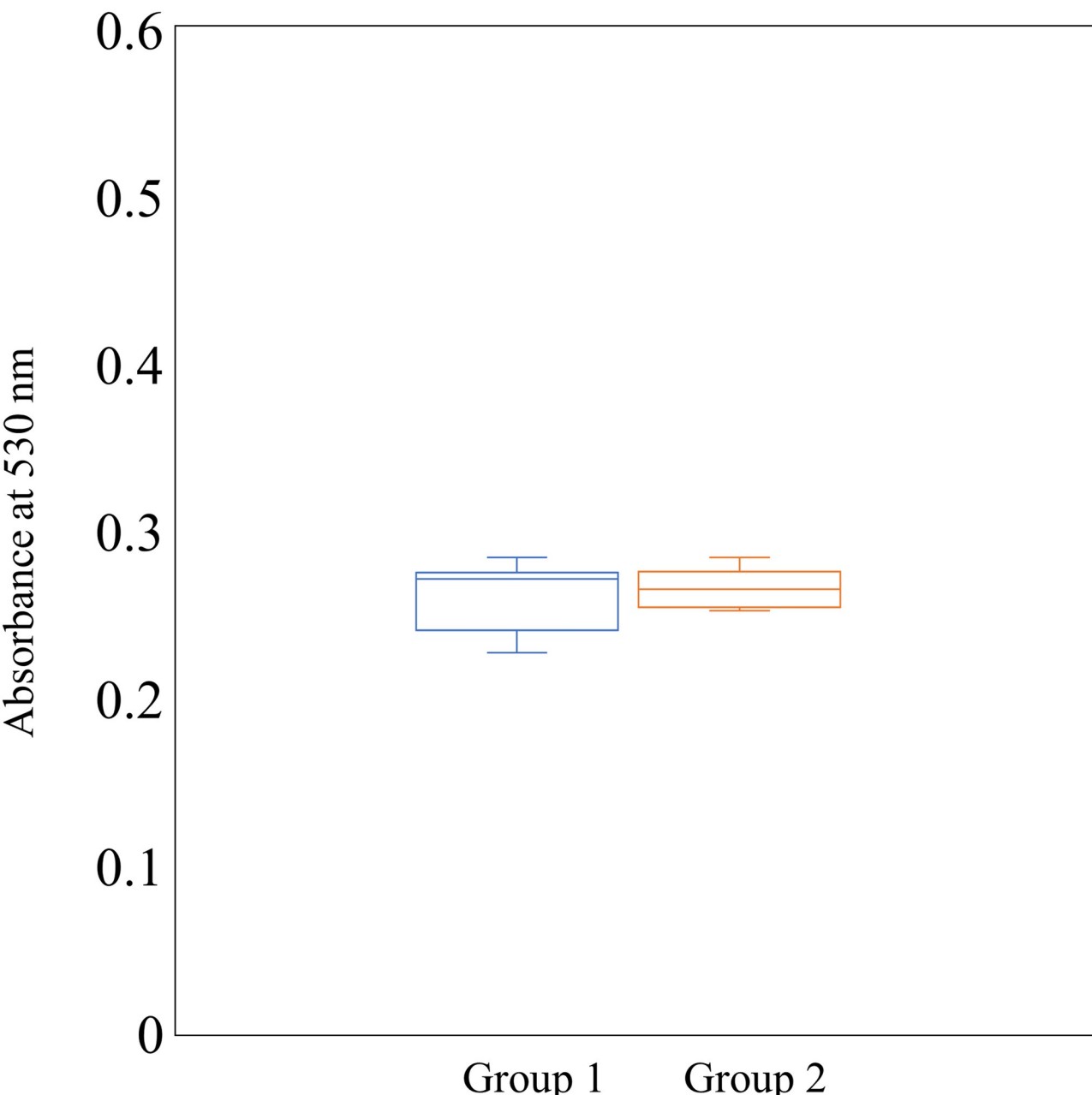

**Fig 6. Cytochrome c binding assay with *C. difficile* strains.** A cytochrome c binding assay with *C. difficile* strains was performed. Student's t test was used to determine significant differences between the Group 1 and Group 2 *C. difficile* strains.

from those in CprA of Group 2 (30 [proline], 70 [serine], 116 [isoleucine] and 193 [lysine]). CprB and CprC in Group 1 also showed differences in 3 and 4 amino acids compared to those in Group 2, respectively.

We further expanded our analysis by using 50 *C. difficile* genomes randomly selected from the NCBI database (Fig 8, S2 and S3 Figs). Although the clusters groups by phylogenetic tree analysis were further divided into several subgroups, the common differences in CprA-C amino acid patterns between Group 1 and Group 2 were also observed in the other genomes that originated somewhere else.

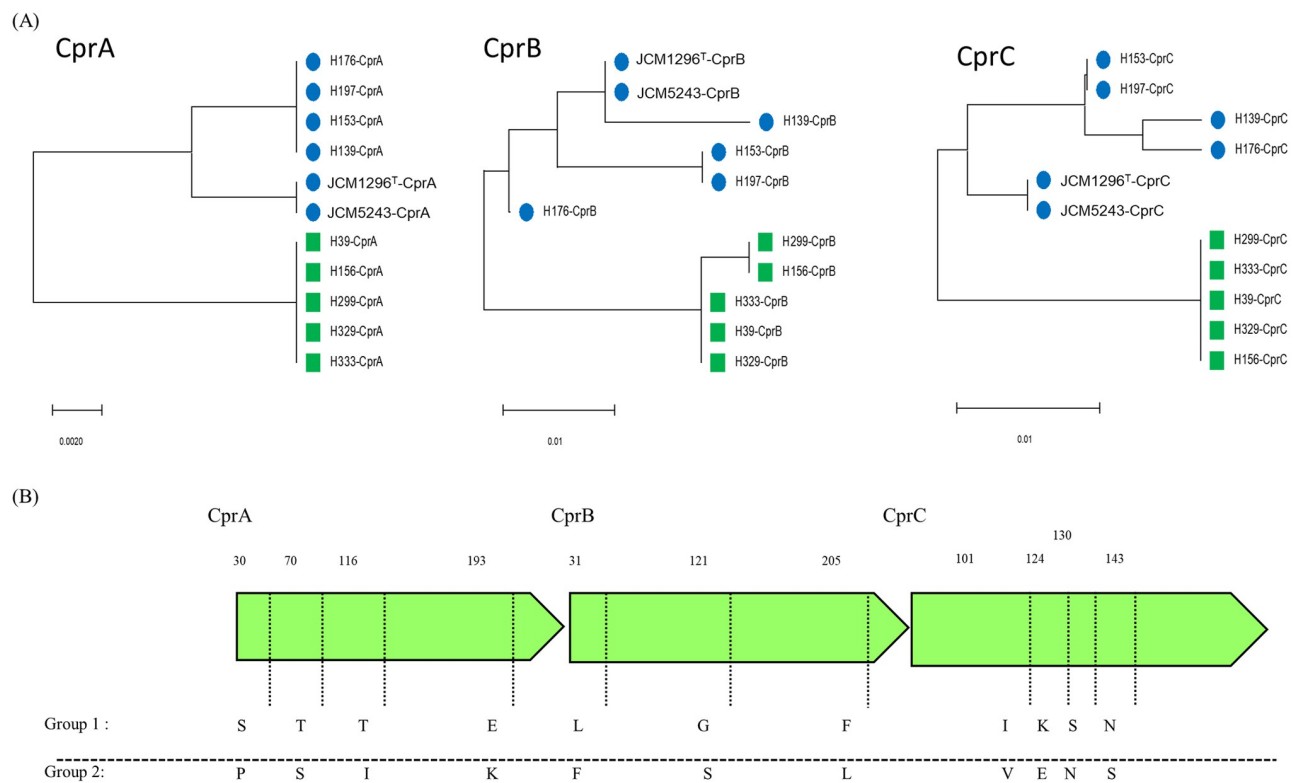

**Fig 7. Comparison of CprABC amino acid sequences among 11 *C. difficile* strains.** (A) Phylogenetic analysis of CprA, B and C among 11 *C. difficile* strains including Group 1 (circle) and Group 2 (square). (B) Conserved amino acid differences between Groups 1 and Group 2.

### Effect of nisin A on *C. difficile* spores and germinated cells

To know the susceptibility to nisin A on spores and germinated cells in Group 1 and 2, we investigated the susceptibility in JCM5243 (Group 1) and H39 (Group 2). In the presence of 0.1% taurocholate, the addition of nisin A (32 or 512 μg/ml) for 30 min drastically reduced the number of cells of both *C. difficile* strains JCM5243 and H39 (Fig 9). In particular, nisin A at 512 μg/ml completely killed the bacterial cells of both strains. In the absence of 0.1% taurocholate, the number of spores in JCM5243 (MIC: 32 μg/ml) in the presence of 32 and 512 μg/ml showed 10.8.0% and 12.5% reduction, respectively, while the number of spores in H39 (MIC: 4 μg/ml) in the presence of 32 and 512 μg/ml showed 24.0% and 87.2% reduction, respectively.

### Discussion

In this study, we found that nisin A susceptibility was different among *C. difficile* strains, which could be divided into 2 groups: high and low nisin A susceptibility. We also found that these groups correlated with the clusters based on whole-genome sequence phylogenetic analysis. Furthermore, it is noteworthy that the amino acid sequences of CprA-C were also divided into 2 groups corresponding to nisin A susceptibility and phylogenetic tree analysis. The *cprA-C* genes encode an ABC transporter, which consists of 2 permeases and 1 ATPase [26, 27]. CprABC was demonstrated to be responsible for resistance to several lantibiotics, including nisin A. *cprK*, encoding histidine kinase, was located downstream of *cprA-C*, while *cprR*, encoding a response regulator, was separate from the *cprA-K* regions. CprKR regulates the expression of *cprA-C* in response to nisin A. We also found that the amino acid sequences of

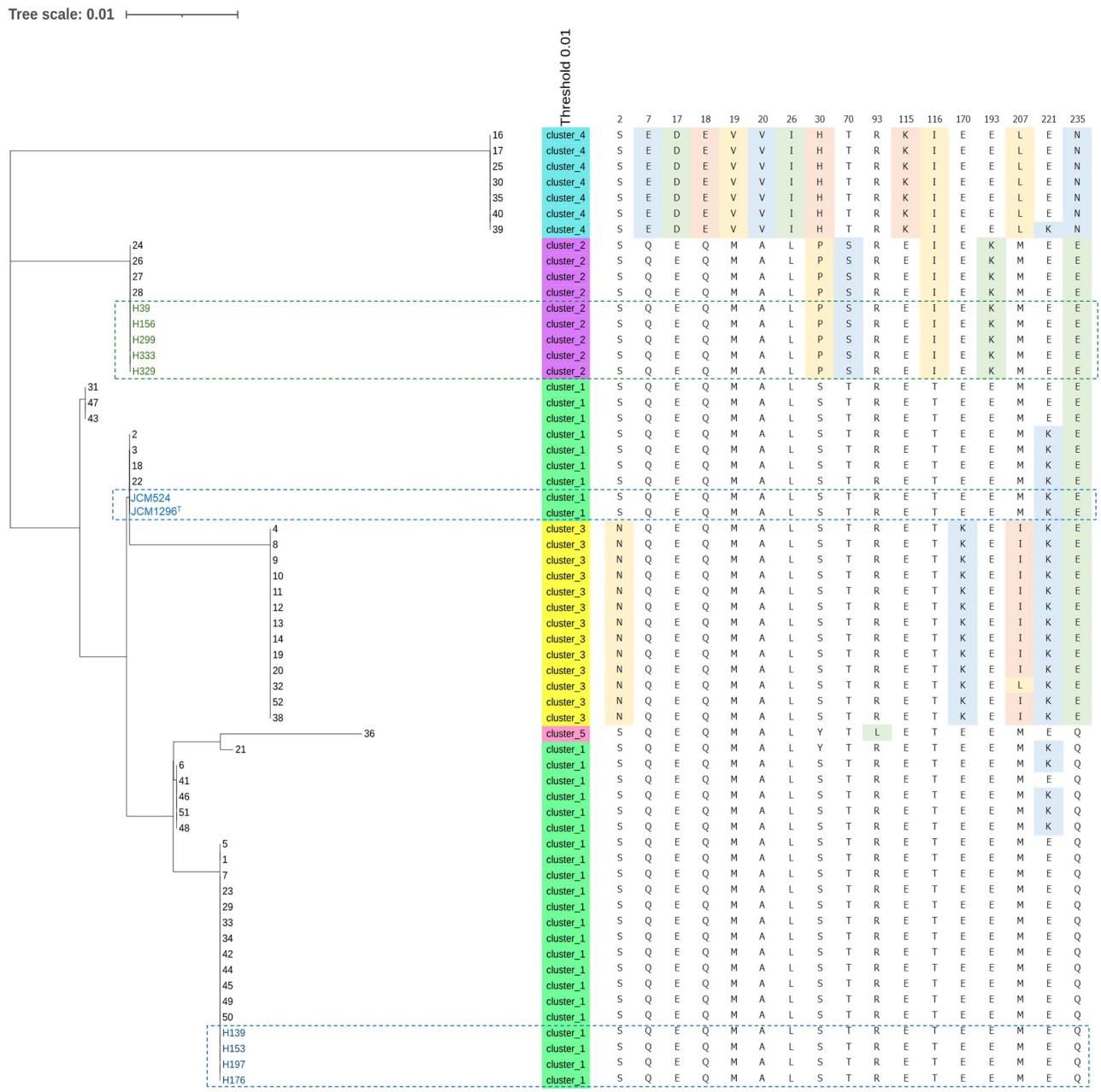

**Fig 8. Phylogenetic analysis of CprA among 61 *C. difficile* strains.** A phylogenetic tree was constructed with 11 the genome sequences determined in this study and 50 sequences obtained from the NCBI database. The blue-dashed square and green-dashed square show strains in Group 1 and Group 2, respectively.

CprK and CprR were divided into 2 groups according to nisin A susceptibility (S4 Fig). However, *cprA* expression in all tested strains was increased by nisin A addition (Fig 5). The expression levels varied among the strains, but were not correlated with nisin A susceptibility. Therefore, we concluded that the different amino acid sequences of CprRK between the 2 groups were not related to nisin A susceptibility.

By comparing the amino acid sequence of CprA-C, we found that several amino acids (4, 3, and 4 amino acid differences in CprA, B and C, respectively) were commonly different in the 2

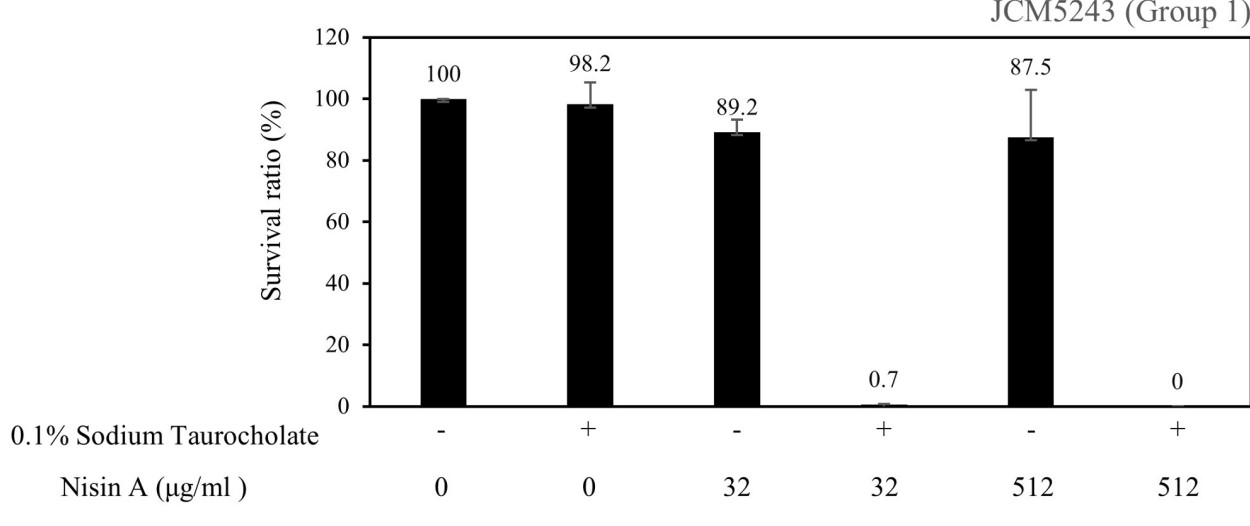

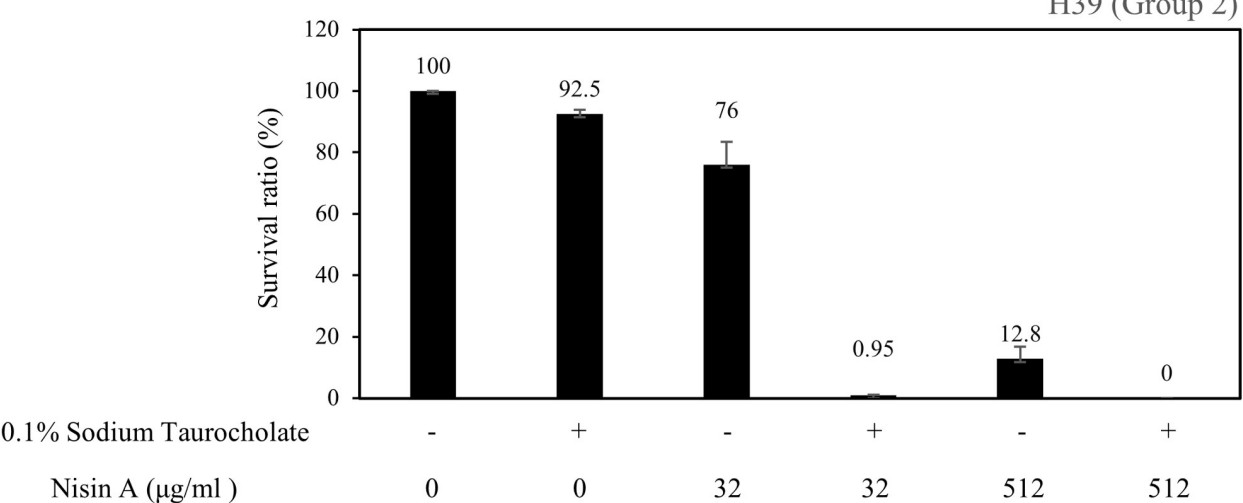

**Fig 9. Effect of nisin A on spores and germinated cells.** The effect of nisin A on the spores and germinated cells of strains JCM5243 and H39. Spores were treated with nisin A (32 or 512 μg/ml) for 30 min in the presence and absence of 0.1% taurocholate.

groups. We further investigated these differences in CprA-C by randomly selecting 50 *C. difficile* genome datasets from the NCBI database. The number of clusters was increased after phylogenetic tree analysis, but these common amino acid differences were still found in each cluster (Fig 8, S2 and S3 Figs). Previously, we found amino acid sequence variations in MutFEG, an ABC transporter responsible for nisin A and mutacin I/III/IIIb resistance in *Streptococcus mutans* [38]. MutFEG was classified into 3 groups (MutFEGs α, β, and γ) based on amino acid sequence. Several amino acid substitutions of MutFEG altered susceptibility to nisin A, mutacin I and mutacin III/IIIb. Based on these results, we suggest that several amino acid differences in CprA-C are associated with susceptibility to nisin A, although we did not directly demonstrate whether these differences affect nisin A susceptibility by replacing *cprA-C* in each type of strain.

Since we found that the *C. difficile* strains separated into 2 groups based on nisin A susceptibility, we further investigated their susceptibility to other bacteriocins, including bacteriocins

with structures similar to nisin A. The first and second ring structures of epidermin and mutacin I/III/IIIb showed a structure similar to that of nisin A [39] (S5 Fig). In particular, the secondary ring structures of epidermin and mutacin III/IIIb were identical to those of nisin A. In Table 3 and Fig 3, we found tendencies of susceptibility to epidermin and mutacin I/III/IIIb similar to that of nisin A, while we did not find the tendencies of susceptibility to class II bacteriocins. Suárez JM demonstrated that the expression of *cprA* was induced by several lantibiotics closely related to nisin A, including gallidermin and mutacin III [26]. From these results, it is speculated that the amino acid differences in CprA-C strongly contributed to susceptibility to nisin A and bacteriocins closely related to nisin A, although their contributions to the level of susceptibility were different.

In addition to CprA-C, the Dlt system in gram-positive bacteria including *C. difficile* has been reported to be involved in resistance to cationic antimicrobial peptides [28, 40–42]. Dlt is involved in alanine incorporation into teichoic or lipoteichoic acid. This incorporation adds a positive charge to these molecules, causing a shift to a weak negative charge on the bacterial cell surface. In this study, we investigated the expression of *dltD* in all strains in the presence and absence of nisin A, and we found that *dltD* expression had no correlation with nisin A susceptibility. In addition, the surface charge evaluated by the cytochrome C binding assay showed no correlation between cell surface charge and nisin A susceptibility. Therefore, we think that DltABCD is not associated with the different susceptibilities to nisin A found in this study.

In this study, we also found no correlation between antibiotic susceptibility and the 2 groups. However, we found that the *vanXYG* genes, which were reported to be involved in vancomycin susceptibility [43], were also only found in type 1 strains, but we did not find a difference in vancomycin susceptibility between the 2 groups. Previous reports also demonstrated that the presence of *vanG* did not contribute to vancomycin resistance [44]. It is interesting to note that the presence of antibiotic resistance genes is linked to *C. difficile* types defined by nisin A susceptibility, although the biological interaction between them remains unknown. Bacteriocin resistance is considered to be important for the survival of bacterial microbiota in certain environments, including the human body. Therefore, type 1 strains might have a strong ability to cause host infection compared to type 2 strains. Furthermore, we investigated the genes for CDT, toxin A (TcdA) and toxin B (TcdB) which are major virulence factors in *C. difficile* [1, 2]. Interestingly, *cdtAB* genes were only found in Group I except H176 strain, while *tcdAB* genes were found in 4 (JCM1296[T], JCM5243, H153 and H197) of 6 strains in Group1 and 2 (H156 and H299) of 5 strains in Group 2. Therefore, the classification of susceptibility by nisin A is also shown to be related to the presence of the gene for CDT.

To know the different susceptibility to nisin A between Group 1 and Group 2 is found in spores or germinated cells, we also investigated the effect of nisin A on spores and germinated cells (Fig 9). Above the MIC of nisin A, germinated cells induced by 0.1% taurocholate were completely killed after 30 min. In contrast, in the absence of 0.1% taurocholate, strain JCM5243, which has low susceptibility to nisin A, did not show a reduction in viable spores after treatment with nisin A (32 or 512 μg/ml) for 30 min, but the H39 strain showed a reduction in viable spores after the addition of nisin A, although the spores were not completely killed. The surface of *C. difficile* spores consists of several layers, including the exosporium, coat, and cortex, which exhibit resistance to many environmental stimuli, such as heat, antibiotics and disinfectants [34]. Therefore, it seems that bacteriocins are not effective against *C. difficile* spores. However, treatment of spores with a high concentration of nisin A in the absence of 0.1% taurocholate reduced the cell (spore) numbers. This result could be explained by the fact that some portions of nisin A attach to the spore surface and show antibacterial activity against the highly susceptible H39 strain after spore germination induced by 0.1% taurocholate.

In conclusion, we first demonstrated the variation in nisin A susceptibility and some lantibiotics that are structurally similar to nisin A. This variation was considered to be due to the differences in CprA-C amino acid sequences that are responsible for nisin A resistance. Moreover, the different amino acid sequences of CprA-C are a result of genomic variations in *C. difficile*. Our results provide a new characteristic feature of *C. difficile* strains and give some considerations for the clinical application of nisin A against *C. difficile* infection.

## Supporting information

**S1 Fig. Susceptibility to nisin A and epidermin against *C. difficile* strains.** Direct assay was performed to evaluate the susceptibility to nisin A and epidermin.
(TIF)

**S2 Fig. Phylogenetic analysis of CprB among 61 *C. difficile* strains.** A phylogenetic tree was constructed with 11 the genome sequences determined in this study and 50 sequences obtained from the NCBI database. The blue-dashed square and green-dashed square show strains in Group 1 and Group 2, respectively.
(TIF)

**S3 Fig. Phylogenetic analysis of CprC among 61 *C. difficile* strains.** A phylogenetic tree was constructed with 11 the genome sequences determined in this study and 50 sequences obtained from the NCBI database. The blue-dashed square and green-dashed square show strains in Group 1 and Group 2, respectively.
(TIF)

**S4 Fig. Comparison of CprKR amino acid sequences among 11 *C. difficile* strains.** Phylogenetic analysis of CprK and CprR among 11 *C. difficile* strains.
(TIF)

**S5 Fig. Structure of nisin A and its structurally related bacteriocins.**
(TIF)

**S1 Table. Primers used in this study.**
(TIF)

## Author Contributions

**Conceptualization:** Miki Kawada-Matsuo.

**Formal analysis:** Noriaki Ide, Miki Kawada-Matsuo, Mi Nguyen-Tra Le, Junzo Hisatsune, Hiromi Nishi, Norikazu Kitamura, Seiya Kashiyama, Hiroki Ohge, Motoyuki Sugai.

**Investigation:** Miki Kawada-Matsuo.

**Project administration:** Miki Kawada-Matsuo.

**Resources:** Toshinori Hara, Michiya Yokozaki, Hiroki Ohge.

**Supervision:** Miki Kawada-Matsuo, Hitoshi Komatsuzawa.

**Validation:** Noriaki Ide, Miki Kawada-Matsuo.

**Visualization:** Noriaki Ide.

**Writing – original draft:** Noriaki Ide, Hiroyuki Kawaguchi.

**Writing – review & editing:** Miki Kawada-Matsuo.

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
