## [Decision Letter · Decision Letter 0]

28 Dec 2022

PONE-D-22-27105Different CprABC amino acid sequences affect nisin A susceptibility in Clostridioides difficile isolatesPLOS ONE

Dear Dr. Komatsuzawa,

Thank you for submitting your manuscript to PLOS ONE. After careful consideration, we feel that it has merit but does not fully meet PLOS ONE’s publication criteria as it currently stands. Therefore, we invite you to submit a revised version of the manuscript that addresses the points raised during the review process.

Please, consider the suggestions of both reviewers, and send the document back for evaluation. 

We look forward to receiving your revised manuscript.

Kind regards,

Guadalupe Virginia Nevárez-Moorillón, Ph.D.

Academic Editor

PLOS ONE

Journal Requirements:

2. Please ensure that you have specified (1) whether consent was informed and (2) what type you obtained (for instance, written or verbal, and if verbal, how it was documented and witnessed). If your study included minors, state whether you obtained consent from parents or guardians. If the need for consent was waived by the ethics committee, please include this information.

3. Please upload a new copy of Figures 4, 5, 7, 8 and 9 as the detail is not clear. Please follow the link for more information: " ext-link-type="uri" xlink:type="simple">https://blogs.plos.org/plos/2019/06/looking-good-tips-for-creating-your-plos-figures-graphics/"
https://blogs.plos.org/plos/2019/06/looking-good-tips-for-creating-your-plos-figures-graphics/

Reviewers' comments:

Reviewer's Responses to Questions

**Comments to the Author**

1. Is the manuscript technically sound, and do the data support the conclusions?

Reviewer #1: Yes

Reviewer #2: Yes

2. Has the statistical analysis been performed appropriately and rigorously? 

Reviewer #1: Yes

Reviewer #2: Yes

3. Have the authors made all data underlying the findings in their manuscript fully available?

Reviewer #1: Yes

Reviewer #2: Yes

4. Is the manuscript presented in an intelligible fashion and written in standard English?

Reviewer #1: Yes

Reviewer #2: Yes

5. Review Comments to the Author

Reviewer #1: The work is well thought out, and carried out the related tests.

The perspectives that the author proposes are appropriate to the results found.

I have brief doubts, which I will write here considering the line number:

Line 18: replace gut microflora with gut microbiota

Line 121: include a “.” after “Table 1”

Line 125: Missing to include the country of Becton Dickinson and Company

Table 1. Why are the recent isolates written as Clostridium, considering that they are now called Clostridioides?

Line 15. Metronidazole should be lowercase in this case.

Line 176: mentions that qPCR, as previously indicated, however, does not include any reference.

Line 182: what is the reason for only using the two indicated strains (JCM5243 and H39)?

Line 190: what was the maximum time before the use of the spores?

Line 225: remove spaces between Table 2 and throughout the text, and check the space between words.

Line 293: unify “Fig.” throughout the text; sometimes, it is not written the same.

Table 4: Is there any parameter according to the CLSI to determine the multi-resistance or susceptibility of C. difficile?

Line 313: from here on, there are values between “( )” bibliographical references that clarify and homogenize the text.

Line 450: replace flora with microbiota

Line 515. Homogenize references; here, the year appears in parentheses, and in the others it does not.

Reviewer #2: In recent years, the problem of drug-resistant bacteria has been the most important issue in the therapeutic treatment of bacterial infections. In the case of Clostridium spp. infection, the occurrence of drug-resistant bacteria has also become serious problem, and the response to this problem needs to be addressed.　In this paper, using 11 clinical isolates of Clostridium difficile, the authors examined the susceptibility to several bacteriocins, including nisin A, and antibiotics. The results of the nisin A susceptibility test showed that the isolates were classified into two major groups, and the classification was similar for the other bacteriocin susceptibility except Mutacin IV. Additionally, no correlation was found between the expression levels of cpr or dlt genes, a group of molecules involved in nisin A resistance, and its susceptibility. Thus, the authors performed a comparative analysis of the whole genome sequences among the isolates to determine the cause of the differences in nisin A susceptibility. As results, the exact same two group formations seen in the susceptibility were confirmed by the genetic information analysis. Furthermore, the authors found that the amino acid sequences of Cpr molecules are unique to each group of bacteria, and speculated that the differences in the amino acid sequences affects the susceptibility to nisinA.

The reviewer considers the current manuscript to be well organized and sufficient for a publication of PLOS ONE if the author will provide suitable responses to the following minor comments;

1. The authors test susceptibility to several bacteriocins in the manuscript. In most bacteriocin susceptibilities, there is a significant difference between the two groups formed by nisinA susceptibility, but not in one. It is strongly expected that this may be caused by structural features of the bacteriocins. However, the current manuscript lacks information on the structure of bacteriocins and their structure-activity relationships. The reviewer recommend that the authors provide a figure showing bacteriocin structure in this regard.

2. Related to comment #1.

What would happen to the susceptibility of isolates if other bacteriocins with less structural homology to nisinA, as like mutacin IV, were used in the test? If the authors have any information on this point, they should provide it or discuss it.

3. The pathogenicity of C. difficile has been well studied, and the typical toxins Tcds and CDT are known to be closely related to its virulence. In this study, the authors have divided the isolates into two groups according to nisinA susceptibility, are there any commonalities in the virulence factors in each group? The authors could add an explanation on this point based on whole genome information, which would further strengthen the value of this paper.

4. The authors need to provide more detailed information regarding the source of the bacteria used in this paper.

5. The authors use taurocholate in the experiments. The purpose of its use should be stated in the Methods section.

6. (Line121) Add period after “Table 1”.

7. (Line176) The authors should add a reference paper regarding qPCR experiment.

6. PLOS authors have the option to publish the peer review history of their article (what does this mean?). If published, this will include your full peer review and any attached files.

Reviewer #1: No

Reviewer #2: No

---

## [Author Response · Author response to Decision Letter 0]

4 Jan 2023

Reviewer #1: The work is well thought out, and carried out the related tests.

The perspectives that the author proposes are appropriate to the results found.

I have brief doubts, which I will write here considering the line number:

Line 18: replace gut microflora with gut microbiota 

Our response: According to the suggestion, we revised it. 

Line 121: include a “.” after “Table 1”

Our response: According to the suggestion, we revised it.

Line 125: Missing to include the country of Becton Dickinson and Company

Our response: According to the suggestion, we revised it.

Table 1. Why are the recent isolates written as Clostridium, considering that they are now called Clostridioides?

Our response: According to the suggestion, we revised it.

Line 15. Metronidazole should be lowercase in this case.

Our response: According to the suggestion, we deleted this sentence.

Line 176: mentions that qPCR, as previously indicated, however, does not include any reference.

Our response: According to the suggestion, we put the reference about qPCR method.

Line 182: what is the reason for only using the two indicated strains (JCM5243 and H39)?

Our response: According to the suggestion, we revised it to explain the reason.

Line 190: what was the maximum time before the use of the spores?

Our response: According to the suggestion, we revised it.

Line 225: remove spaces between Table 2 and throughout the text, and check the space between words.

Our response: According to the suggestion, we revised it.

Line 293: unify “Fig.” throughout the text; sometimes, it is not written the same.

Our response: According to the suggestion, we revised it throughout the text.

Table 4: Is there any parameter according to the CLSI to determine the multi-resistance or susceptibility of C. difficile?

Our Response: There are MIC breakpoints of several antibiotics according to CLSI. However, among antibiotics used in this study, vancomycin and metronidazole are included in the MIC breakpoints of C. difficile by CLSI, but other antibiotics are not included. However, we wanted to compare the MIC value between 2 groups and found no correlation between 2 groups.

Line 313: from here on, there are values between “( )” bibliographical references that clarify and homogenize the text.

Our response: According to the suggestion, we revised it.

Line 450: replace flora with microbiota

Our response: According to the suggestion, we revised it.

Line 515. Homogenize references; here, the year appears in parentheses, and in the others it does not.

Our response: According to the suggestion, we revised it.

Reviewer #2: In recent years, the problem of drug-resistant bacteria has been the most important issue in the therapeutic treatment of bacterial infections. In the case of Clostridium spp. infection, the occurrence of drug-resistant bacteria has also become serious problem, and the response to this problem needs to be addressed.　In this paper, using 11 clinical isolates of Clostridium difficile, the authors examined the susceptibility to several bacteriocins, including nisin A, and antibiotics. The results of the nisin A susceptibility test showed that the isolates were classified into two major groups, and the classification was similar for the other bacteriocin susceptibility except Mutacin IV. Additionally, no correlation was found between the expression levels of cpr or dlt genes, a group of molecules involved in nisin A resistance, and its susceptibility. Thus, the authors performed a comparative analysis of the whole genome sequences among the isolates to determine the cause of the differences in nisin A susceptibility. As results, the exact same two group formations seen in the susceptibility were confirmed by the genetic information analysis. Furthermore, the authors found that the amino acid sequences of Cpr molecules are unique to each group of bacteria, and speculated that the differences in the amino acid sequences affects the susceptibility to nisinA.

The reviewer considers the current manuscript to be well organized and sufficient for a publication of PLOS ONE if the author will provide suitable responses to the following minor comments;

1.The authors test susceptibility to several bacteriocins in the manuscript. In most bacteriocin susceptibilities, there is a significant difference between the two groups formed by nisinA susceptibility, but not in one. It is strongly expected that this may be caused by structural features of the bacteriocins. However, the current manuscript lacks information on the structure of bacteriocins and their structure-activity relationships. The reviewer recommend that the authors provide a figure showing bacteriocin structure in this regard.

Our responses: According to the suggestion, we showed the structure of bacteriocins including nisin A and its structural related bacteriocins which were discussed in Discussion section as the supplemental figure (Supplemental Fig. 5). 

2. Related to comment #1.

What would happen to the susceptibility of isolates if other bacteriocins with less structural homology to nisinA, as like mutacin IV, were used in the test? If the authors have any information on this point, they should provide it or discuss it.

Our responses: According to the suggestion, we investigated the susceptibility of other bacteriocins with no structural relation to nisin A. As the result, we found no tendency as nisin A susceptibility. Therefore, the different amino acid sequences of CprA-C only affected the susceptibility of nisin A and its-related bacteriocins. We revised Table 3 and Fig. 3 by adding two bacteriocins, and also discussed it in Discussion section.

3. The pathogenicity of C. difficile has been well studied, and the typical toxins Tcds and CDT are known to be closely related to its virulence. In this study, the authors have divided the isolates into two groups according to nisinA susceptibility, are there any commonalities in the virulence factors in each group? The authors could add an explanation on this point based on whole genome information, which would further strengthen the value of this paper.

Our responses: We investigated the genes coding for Cdt and Txoin AB (TcdAB), and found a tendency of cdt genes in Group I and II, but did not find the tendency of tcdAB. We added these results in Discussion section. 

4. The authors need to provide more detailed information regarding the source of the bacteria used in this paper.

Our responses: According to the suggestion, we added the detailed information of the strains in Methods section. Also, the strain name of JCM1296 was corrected to JCM1296T.

5. The authors use taurocholate in the experiments. The purpose of its use should be stated in the Methods section.

Our responses: According to the suggestion, we describe the purpose of taurocholate in Material and Methods section.

6. (Line121) Add period after “Table 1”.

Our responses: According to the suggestion, we revised it.

7. (Line176) The authors should add a reference paper regarding qPCR experiment.

Our responses: According to the suggestion, we put the reference.

---

## [Editor Report · Decision Letter 1]

6 Jan 2023

Different CprABC amino acid sequences affect nisin A susceptibility in  Clostridioides difficile isolates

PONE-D-22-27105R1

Dear Dr. Komatsuzawa,

We’re pleased to inform you that your manuscript has been judged scientifically suitable for publication and will be formally accepted for publication once it meets all outstanding technical requirements.

Kind regards,

Guadalupe Virginia Nevárez-Moorillón, Ph.D.

Academic Editor

PLOS ONE
---

## [Editor Report · Acceptance letter]

12 Jan 2023

PONE-D-22-27105R1 

Different CprABC amino acid sequences affect nisin A susceptibility in *Clostridioides difficile* isolates 

Dear Dr. Komatsuzawa:

I'm pleased to inform you that your manuscript has been deemed suitable for publication in PLOS ONE. Congratulations! Your manuscript is now with our production department. 

Kind regards, 

on behalf of

Dr. Guadalupe Virginia Nevárez-Moorillón 

Academic Editor

PLOS ONE